# BIAS ALSO MATTERS: BIAS ATTRIBUTION FOR DEEP NEURAL NETWORK EXPLANATION

## ABSTRACT

The gradient of a deep neural network (DNN) w.r.t. the input provides information that can be used to explain the output prediction in terms of the input features and has been widely studied to assist in interpreting DNNs. In a linear model (i.e., $g(x) = wx + b$), the gradient corresponds solely to the weights $w$. Such a model can reasonably locally linearly approximate a smooth nonlinear DNN, and hence the weights of this local model are the gradient. The other part, however, of a local linear model, i.e., the bias $b$, is usually overlooked in attribution methods since it is not part of the gradient. In this paper, we observe that since the bias in a DNN also has a non-negligible contribution to the correctness of predictions, it can also play a significant role in understanding DNN behaviors. In particular, we study how to attribute a DNN's bias to its input features. We propose a backpropagation-type algorithm "bias back-propagation (BBp)" that starts at the output layer and iteratively attributes the bias of each layer to its input nodes as well as combining the resulting bias term of the previous layer. This process stops at the input layer, where summing up the attributions over all the input features exactly recovers $b$. Together with the backpropagation of the gradient generating $w$, we can fully recover the locally linear model $g(x) = wx + b$. Hence, the attribution of the DNN outputs to its inputs is decomposed into two parts, the gradient $w$ and the bias attribution, providing separate and complementary explanations. We study several possible attribution methods applied to the bias of each layer in BBp . In experiments, we show that BBp can generate complementary and highly interpretable explanations of DNNs in addition to gradient-based attributions.

## 1 INTRODUCTION

Deep neural networks (DNNs) have produced good results for many challenging problems in computer vision, natural language processing, and speech processing. Deep learning models, however, are usually designed using fairly high-level architectural decisions, leading to a final model that is often seen as a difficult to interpret black box. DNNs are a highly expressive trainable class of non-linear functions, utilizing multi-layer architectures and a rich set of possible hidden non-linearities, making interpretation by a human difficult. This restricts the reliability and usability of DNNs especially in mission-critical applications where a good understanding of the model's behavior is necessary.

The gradient is a useful starting point for understanding and generating explanations for the behavior of a complex DNN. Having the same dimension as the input data, the gradient can reflect the contribution to the DNN output of each input dimension. Not only does the gradient yield attribution information for every data point, but also it helps us understand other aspects of DNNs, such as the highly celebrated adversarial examples and defense methods against such attacks (Szegedy et al., 2013).

When a model is linear, the gradient recovers the weight vector. Since a linear model locally approximates any sufficiently smooth non-linear model, the gradient can also be seen as the weight vector of that local linear model for a given DNN at a given data point. For a piecewise linear DNN (e.g., a DNN with activation functions such as ReLU, LeakyReLU, PReLU, and hard tanh) the gradient is exactly the weights of the local linear model[1].

---

[1]This is true except when the gradient is evaluated at an input on the boundary of the polyhedral region within which the DNN equals to the local linear model. In such case, subgradients are appropriate.

Although the gradient of a DNN has been shown to be helpful in understanding the behavior of a DNN, the other part of the locally linear model, i.e., the bias term, to the best of our knowledge, has not been studied explicitly and is often overlooked. If only considering one linear model within a small region, the bias, as a scalar, seems to contain less information than the weight vector. However, this scalar is the result of complicated processing of bias terms over every neuron and every layer based on the activations, the non-linearity functions, as well as the weight matrices of the network. Uncovering the bias's nature could potentially reveal a rich vein of attribution information complementary to the gradient. For classification tasks, it can be the case that the gradient part of the linear model contributes to only a negligible portion of the target label's output probability (or even a negative logit value), and only with a large bias term does the target label's probability becomes larger than that of other labels to result in the correct prediction (see Sec 5). In our empirical experiments (Table 1), using only the bias term of the local linear models achieves 30-40% of the performance of the complete DNN, thus indicating that the bias term indeed plays a substantial role in the mechanisms of a DNN.

In this paper, we unveil the information embedded in the bias term by developing a general bias attribution framework that distributes the bias scalar to every dimension of the input data. We propose a backpropagation-type algorithm called "bias backpropagation (BBp)" to send and compute the bias attribution from the output and higher-layer nodes to lower-layer nodes and eventually to the input features, in a layer-by-layer manner. Specifically, BBp utilizes a recursive rule to assign the bias attribution on each node of layer $\ell$ to all the nodes on layer $\ell - 1$, while the bias attribution on each node of layer $\ell - 1$ is composed of the attribution sent from the layer below and the bias term incurred in layer $\ell - 1$. The sum of the attributions over all input dimensions produced by BBp exactly recovers the bias term in the local linear model representation of the DNN at the given input point. In experiments, we visualize the bias attribution results as images on a DNN trained for image classification. We show that bias attribution can highlight essential features that are complementary from what the gradient-alone attribution methods favor.

## 2 RELATED WORK

Attribution methods for deep models is an important modern research in machine learning since it is important to complement the good empirical performance of DNNs with explanations for how, why, and in what manner do such complicated models make their decisions. Ideally, such methods would render DNNs to be glass boxes rather than black boxes. To this end, a number of strategies have been investigated. Simonyan et al. (2013) visualized behaviors of convolutional networks by investigating the gradients of the predicted class output with respect to the input features. Deconvolution (Zeiler & Fergus, 2014) and guided backpropagation (Springenberg et al., 2014) modify gradients with additional constraints. Montavon et al. (2017) extended to higher order gradient information by calculating the Taylor expansion, and Binder et al. (2016) study the Taylor expansion approach on DNNs with local renormalization layers. Shrikumar et al. (2017) proposed DeepLift, which separated the positive attribution and negative attribution, and featured customer designed attribution scores. Sundararajan et al. (2017) declared two axioms an attribution method needs to satisfy. It further developed an integrated gradient method that accumulates gradients on a straightline path from a base input to a real data point and uses the aggregated gradients to measure the importance of input features. Class Activation Mapping (CAM) (Zhou et al., 2016) localizes the attribution based on the activation of convolution filters, and can only be applied to a fully convolutional network. Grad-CAM (Selvaraju et al., 2017) relaxes the all-convolution constraints of CAM by incorporating the gradient information from the non-convolutional layers. All the work mentioned above utilizes information encoded in the gradients in some form or another, but none of them explicitly investigates the importance of the bias terms, which is the focus of this paper. Some of them, e.g. Shrikumar et al. (2017) and Sundararajan et al. (2017), consider the overall activation of neurons in their attribution methods, so the bias terms are implicitly taken into account, but are not independently studied. Moreover, some of the previous work (e.g. CAM) focus on the attribution for specific network architectures such as convolutional networks, while our approach generally applies to any piece-wise linear DNN, convolutional or otherwise.

## 3 BACKGROUND AND MOTIVATION

We can write the output $f(x)$ of any feed-forward deep neural network in the following form:
$$f(x) = W_m \psi_{m-1}(W_{m-1}\psi_{m-2}(\dots \psi_1(W_1 x + b_1) \dots) + b_{m-1}) + b_m. \tag{1}$$
Where $W_i$ and $b_i$ are the weight matrix and bias term for layer $i$, $\psi_i$ is the corresponding activation function, $x \in X$ is an input data point of $d_{in}$ dimensions, $f(x)$ is the network's output prediction

of $d_{out}$ dimensions, and each hidden layer $i$ has $d_i$ nodes. In this paper, we rule out the last softmax layer from the network structure; for example, the output $f(x)$ may refer to logits (which are the inputs to a softmax to compute probabilities) if the DNN is trained for classification tasks.

The above formalization of DNN generalizes many widely used DNN architectures. Clearly, Eq. ((1)) can represent a fully-connected network of $m$ layers. Moreover, the convolution operation is essentially a matrix multiplication, where every row of the matrix corresponds to applying a filter from convolution on a certain part of the input, and therefore the resulting weight matrix has tied parameters and is very sparse, and typically has a very large (compared to the input size) number of rows. The average-pooling is essentially a linear operation and therefore is representable as a matrix multiplication, and max-pooling can be treated as an activation function. Batchnorm (Ioffe & Szegedy, 2015) is a linear operation and can be combined into the weight matrix. Finally, we can represent a residual network (He et al., 2015) block by appending an identity matrix at the bottom of a weight matrix so that we can keep the input values, and then add the kept input values later through another matrix operation.

## 3.1 PIECEWISE LINEAR DEEP NEURAL NETWORKS

In this paper, we will focus on DNNs with piecewise linear activation functions, which cover most of the successfully used neural networks in a variety of application domains. Some widely used piecewise linear activation functions include the ReLU, leaky ReLU, PReLU, and the hard tanh functions. A general form of a piecewise linear activation function applied to a real value $z$ is as follows:

$$\psi(z) = \begin{cases} c^{(0)} \cdot z, & \text{if } z \in (\eta_0, \eta_1] \\ c^{(1)} \cdot z, & \text{if } z \in (\eta_1, \eta_2] \\ \cdots, & \cdots \\ c^{(h-1)} \cdot z, & \text{if } z \in (\eta_{h-1}, \eta_h) \end{cases} \tag{2}$$

In the above, there are $h$ linear pieces, and these correspond to $h$ predefined intervals on the real axis. We define the activation pattern $\phi(z)$ of $z$ as the index of the interval containing $z$, which can be any integer from 0 to $h - 1$. Both $\psi(z)$ and $\phi(z)$ extend to element-wise operators when applied to vectors or high dimensional tensors.

As long as the activation function is piecewise linear, the DNN is a piecewise linear function and is equivalent to a linear model at each input point $x$. Specifically, each piece of the DNN (associated with an input point $x$) is a linear model:

$$f(x) = W_m^x W_{m-1}^x \cdots W_1^x x + (W_m^x W_{m-1}^x \cdots W_2^x b_1^x + \cdots + W_m^x W_{m-1}^x b_{m-2}^x + W_m^x b_{m-1}^x + b_m)$$

$$= \prod_{i=1}^m W_i^x x + \left( \sum_{j=2}^m \prod_{i=j}^m W_i^x b_{j-1}^x + b_m \right) = \frac{\partial f(x)}{\partial x} x + b^x. \tag{3}$$

This holds true for all the possible input points $x$ on the linear piece of DNN. We will give a more general result later in Lemma 1. Note $W_i^x$ and $b_i^x$ in the above linear model are modified from $W_i$ and $b_i$ respectively and have to fulfill

$$x_{i+1} = \psi_i(W_i x_i + b_i) = W_i^x x_i + b_i^x, \tag{4}$$

where $x_i$ is the activation of layer $i$ ($x_1$ is the input data) and $b_i^x$ is an $x_i$-dependent bias vector vector. In the extreme case, no two input training data points share the same linear model in Eq. (3). Note in this case, the DNN can still be represented as a piecewise linear model on a set of input data points, and each local linear model is only applied to one data point.

Given $x_i$, $W_i^x$ and $b_i^x$ can be derived from $W_i$ and $b_i$ according to the activation pattern vector $\phi(W_i x_i + b_i)$. In particular, each row of $W_i^x$ is a scaled version of the associated row of $W_i$, and each element in $b_i^x$ is a scaled version of $b_i$, i.e.,

$$W_i^x[p] = c^{(\phi(W_i x_i + b_i)[p])} \cdot W_i, \quad \text{and} \quad b_i^x[p] = c^{(\phi(W_i x_i + b_i)[p])} \cdot b_i. \tag{5}$$

For instance, if ReLU $\psi_{ReLU}(z) = \max(0, z)$ is used as the activation function $\psi(\cdot)$ at every layer $i$, we have an activation pattern $\phi(W_i x_i + b_i) \in \{0, 1\}^{d_i}$, where $\phi(W_i x_i + b_i)[p] = 0$ indicates that ReLU sets the output node $p$ to 0 or otherwise preserves the node value. Therefore, at layer $i$, $W_i^x$ and $b_i^x$ are modified from $W_i$ and $b_i$ by setting the rows of $W_i$, whose corresponding activation patterns in $\phi(W_i x_i + b_i)$ are 0, to be all-zero vectors $\mathbf{0}$, and setting the associated elements in $b_i^x$ to be 0 while other elements to be $b_i$.

We can apply the above process to deeper layers as well, eliminating all the ReLU functions to produce an $x$-specific local linear model representing one piece of the DNN, as shown in Eq. (3). Since the model is linear, the gradient $\frac{\partial f(x)}{\partial x}$ is the weight vector of the linear model. Also, given

all the weights of the DNN, each linear region, and the associated linear model can be uniquely determined by the ReLU patterns $\{\phi(x_i)\}_{i=2}^m$, which are $m$ binary vectors.

## 3.2 ATTRIBUTION OF DNN OUTPUTS TO INPUTS

Given a specific input point $x$, the attribution of each dimension $f(x)[j]$ of the DNN output (e.g., the logit for class $j$) to the input features aims to assign a portion of $f(x)[j]$ to each of the input features $i$, and all the portions assigned to all the input features should sum up to $f(x)[j]$. For simplicity, in the rest of this paper, we rename $f(x)$ to be $f(x)[j]$, which does not lose any generality since the same attribution method can be applied for any output dimension $j$. According to Eq. (3), $f(x)$ as a linear model on $x$ can be decomposed into two parts, the linear transformation $\frac{\partial f(x)}{\partial x}$ and the bias term $b^x$. The attribution of the first part is straightforward because we can directly assign each dimension of the gradient $\frac{\partial f(x)}{\partial x}$ to the associated input feature, and we can generate the gradient by using the standard backpropagation algorithm. The gradient-based attribution methods have been widely studied in previous work (see Section 2). However, the attribution of the second part, i.e., the bias $b$, is arguably a more challenging problem since it is not obvious how to assign a portion of $b$ to each input feature since $b$ is a scalar value rather than a vector that, like the gradient, has the same dimensionality as the input vector.

One possible reason for the dearth of bias attribution studies might be that people consider bias, as a scalar, less important relative to the weight vector, containing only minor information about deep model decisions. The final bias scalar $b^x$ of every local linear model, however, is the result of a complex process (see Eq. (3)), where the bias term on every neuron of a layer gets modified based on the activation function (e.g., for ReLU, a bias term gets dropped if the neuron has a negative value), then propagates to the next layer based on the weight matrix, and contribute to the patterns of activation function in the next layer. As the bias term applied to every neuron can be critical in determining the activation pattern (e.g., changing a neuron output from negative to positive for ReLU), we wish to be able to better understand the behavior of deep models by unveiling and reversing the process of how the final bias term is generated.

Moreover, as we show in our empirical studies (see Section 5), we train DNNs both with and without bias for image classification tasks, and the results show that the bias plays a significant role in producing accurate predictions. In fact, we find that it is not rare that the main component of a final logit, leading to the final predicted label, comes from the bias term, while the gradient term $\frac{\partial f(x)}{\partial x}x$ makes only a minor, or even negative, contribution to the ultimate decision. In such a case, ignoring the bias term can provide misleading input feature attributions.

Intuitively, the bias component also changes the geometric shape of the piecewise linear DNNs (see Fig. 1); this means that it is an essential component of deep models and should also be studied, as we do in this paper.

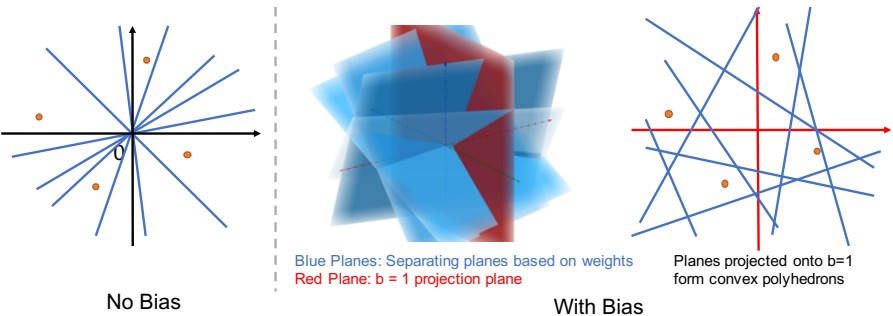

Blue Planes: Separating planes based on weights
Red Plane: b = 1 projection plane

Planes projected onto b=1 form convex polyhedrons

No Bias                                 With Bias

Figure 1: A Piecewise linear weight matrix divides the input plane into regions. Without the bias term, the regions are cones, while with the bias term, the regions are convex polyhedra.

It is a mathematical fact that a piecewise linear DNN is equivalent to a linear model for each input data point. Therefore, the interpretation of the DNN's behavior on the input data should be exclusive to the information embedded in the linear model. However, we often find that the gradient of the DNN, or the weight of the linear model, does not always produce satisfying explanations in practice, and in many cases, it may be due to the overlooked attribution of the bias term that contains the complementary or even key information to make the attribution complete.

## 4 BIAS BACKPROPAGATION FOR BIAS ATTRIBUTION

In this section, we will introduce our method for bias attribution. In particular, the goal is to find a vector $\beta$ of the same dimension $d_{in}$ as the input data point $x$ such that $\sum_{p=1}^{d_{in}} \beta[p] = b^x$. However, it is not clear how to directly assign a scalar value $b$ to the $d_{in}$ input dimensions, since there are $m$ layers between the outputs and inputs. In the following, we explore the neural net structure for bias attribution and develop a backpropagation-type algorithm to attribute the bias $b$ layer by layer from the output $f(x)$ to the inputs in a bottom-up manner.

### 4.1 BIAS BACKPROPAGATION (BBP)

Recall $x_\ell$ denotes the input nodes of layer $\ell \geq 2$, i.e.,
$$x_\ell = \psi_{\ell-1}(W_{\ell-1}x_{\ell-1} + b_{\ell-1}) = \psi_{\ell-1}(W_{\ell-1}\psi_{\ell-2}(\ldots\psi_1(W_1 x + b_1)\ldots) + b_{\ell-1}). \quad (6)$$
According to the recursive computation shown in Eq. (3), the output $f(x)$ can be represented as a linear model of $x_\ell$ shown in the following lemma.

**Lemma 1.** *Given $x$, the output $f(x)$ of a piecewise linear DNN can be written as a linear model of the input $x_\ell$ of any layer $\ell > 2$ ($x_1 = x$ is the raw input) in the following form.*

$$f(x) = \left(\prod_{i=\ell}^{m} W_i^x\right) x_\ell + \left(\sum_{j=\ell+1}^{m}\prod_{i=j}^{m} W_i^x b_{j-1}^x + b_m\right). \quad (7)$$

For each input node $x_\ell[p]$ of layer $\ell$, we aim to compute $\beta_\ell[p]$ as the bias attribution on $x_\ell[p]$. We further require that summing $\beta_\ell[p]$ over all input nodes of layer $\ell$ recovers the bias in Eq. (7), i.e.,

$$\sum_{p=1}^{d_\ell} \beta_\ell[p] = \sum_{j=\ell+1}^{m}\prod_{i=j}^{m} W_i^x b_{j-1}^x + b_m, \quad (8)$$

so the linear model in Eq. (7) can be represented as the sum of $d_\ell$ terms associated with the $d_\ell$ input nodes, each composed of a linear transformation part and a bias attribution part, i.e.,

$$f(x) = \sum_{p=1}^{d_\ell} \left[\left(\prod_{i=\ell}^{m} W_i^x\right)[p] \cdot x_\ell[p] + \beta_\ell[p]\right]. \quad (9)$$

Note the above equation gives the attribution of the output $f(x)$ on each hidden node $x_\ell[p]$ of the DNN. It is composed of two parts, i.e., the gradient attribution and the bias attribution. Since the bias in the right-hand side of Eq. (8) can be represented as an accumulated sum of the bias terms incurred from the last layer to layer $\ell$ (i.e., $b_m$ for the last layer and $\prod_{i=j}^{m} W_i^x b_{j-1}^x$ for layer $j-1$), we can design a recursive rule that computes the bias attribution on layer $\ell-1$ given the bias attribution $\beta_\ell$ on layer $\ell$. In particular, we assign different portions of $\beta_\ell[p]$ to each node $x_{\ell-1}[q]$ on layer $\ell-1$, and make sure that summing up those portions recovers $\beta_\ell[p]$. Each portion $B_\ell[p,q]$ can be treated as a message regarding bias attribution that node $x_\ell[p]$ sends to node $x_{\ell-1}[q]$. For each node $x_\ell[p]$ of layer $\ell$, we compute a vector of attribution scores $\alpha_\ell[p]$, and define the message $B_\ell[p,q]$ as

$$B_\ell[p,q] \triangleq \alpha_\ell[p,q] \times \beta_\ell[p], \quad \sum_{q=1}^{d_{\ell-1}} \alpha_\ell[p,q] = 1 \text{ and, } \forall p \in [d_\ell], q \in [d_{\ell-1}]. \quad (10)$$

We will discuss several options to compute the attribution scores $\alpha_\ell[p]$ later. To make our bias attribution method flexible and compatible with any attribution function, we allow both negative scores and positive scores in $\alpha_\ell[p]$.

The bias attribution $\beta_{\ell-1}[q]$ on node $x_{\ell-1}[q]$ of layer $\ell-1$ is achieved by firstly summing up the bias attribution messages sent from nodes in layer $\ell$, and then adding the bias term $\prod_{i=\ell}^{m} W_i^x b_{j-1}^x$ incurred in layer $\ell-1$ (which is applied to all nodes in layer $\ell-1$), as shown below:

$$\beta_{\ell-1}[q] = \prod_{i=\ell}^{m} W_i^x b_{j-1}^x + \sum_{p=1}^{d_\ell} B_\ell[p,q]. \quad (11)$$

It can be easily verified that summing up the attribution $\beta_{\ell-1}[q]$ over all the nodes on layer $\ell-1$ yields the bias term in Lemma 1, when writing $f(x)$ at $x$ as a linear model of $x_{\ell-1}$, i.e.,

$$\sum_{q=1}^{d_{\ell-1}} \beta_{\ell-1}[q] = \sum_{j=\ell}^{m}\prod_{i=j}^{m} W_i^x b_{j-1}^x + b_m. \quad (12)$$

Hence, the complete attribution of $f(x)$ on the nodes of layer $\ell - 1$ can be written in the same form as the one shown in Eq. (9) for layer $\ell$, i.e., $f(x) = \sum_{q=1}^{d_{\ell-1}} \left[ \left( \prod_{i=\ell-1}^{m} W_i^x \right) [q] \cdot x_{\ell-1}[q] + \beta_{\ell-1}[q] \right]$. Therefore, we start from the last layer, and recursively apply Eq. (10)-(11) from the last layer to the first layer. This process backpropagates to the lower layers the bias term incurred in each layer and the bias attributions sent from higher layers. Eventually, we can obtain the bias attribution $\beta[p]$ for each input dimension $p$. The bias attribution algorithm is detailed in Algorithm 1.

---

**Algorithm 1:** Bias Backpropagation (BBp)

**input** : $x, \{W_\ell\}_{\ell=1}^m, \{b_\ell\}_{\ell=1}^m, \{\psi_\ell(\cdot)\}_{\ell=1}^m$
1  Compute $\{W_\ell^x\}_{\ell=1}^m$ and $\{b_\ell^x\}_{\ell=1}^m$ for $x$ by Eq. (5) ;          // Get data point specific weight/bias
2  $\beta_m \leftarrow b_m$ ;          // $\beta_\ell$ holds the accumulated attribution for layer $\ell$
3  **for** $\ell \leftarrow m$ **to** 2 **by** $-1$ **do**
4      **for** $p \leftarrow 1$ **to** $d_\ell$ **by** 1 **do**
5          Compute $\alpha_\ell[p]$ by Eq. (13)-(15) or Eq. (16) ;          // Compute attribution score
6          $B_\ell[p, q] \leftarrow \alpha_\ell[p, q] \times \beta_\ell[p], \ \forall q \in [d_{\ell-1}]$ ;          // Attribute to the layer input
7      **end**
8      **for** $q \leftarrow 1$ **to** $d_{\ell-1}$ **by** 1 **do**
9          $\beta_{\ell-1}[q] \leftarrow \prod_{i=\ell}^{m} W_i^x b_{j-1}^x + \sum_{p=1}^{d_\ell} B_\ell[p, q]$ ;          // Combine with bias of layer $\ell - 1$
10     **end**
11 **end**
12 **return** $\beta_1 \in \mathbb{R}^{d_{in}}$ ;

---

### 4.2 Options to Compute Attribution Scores in $\alpha_\ell[p]$

In the following, we discuss two possible options to compute the attribution scores in $\alpha_\ell[p]$, where $\alpha_\ell[p, q]$ measures how much of the bias $x_\ell[p]$ should be attributed to $x_{\ell-1}[q]$. For both of options, we design $\alpha_\ell[p]$ so that the bias attribution on each neuron serves as a compensation for the weight or gradient term to achieve the desired output value, and thus the bias attribution can give complementary explanations to the gradient attribution.

We have $x_\ell[p] = \sum_{r:=1}^{d_{l-1}} W_{\ell-1}^x[p, r] x_{\ell-1}[r] + b_\ell^x[p]$. Suppose $b_\ell^x[p]$ is negative, we may reason that to achieve the target value of $x_\ell[p]$, the positive components of the gradient term $\sum_{r:=1}^{d_{l-1}} W_{\ell-1}^x[p, r] x_{\ell-1}[r]$ are larger than desirable, so that we need to apply the additional negative bias in order to achieve the desired output $x_\ell[p]$. In other words, the large positive components can be thought as the causal factor leading to the negative bias term, so we attribute more bias to the larger positive components.

On the other hand, suppose $b_\ell^x[p]$ is positive, then the negative components of the gradient term are smaller (or larger in magnitude) than desirable, so the small negative values cause the bias term to be positive, and therefore, we attribute more bias to the smaller negative components. Thus, we have

$$\alpha_\ell[p, q] = \frac{\mathbb{1}_{e(l-1,p,q)=1} \exp(s_\ell[p, q]/T)}{\sum_{r=1}^{d_{\ell-1}} \mathbb{1}_{e(l-1,p,r)=1} \exp(s_\ell[p, r]/T)}, \tag{13}$$

$$\text{where } s_\ell[p, q] = -\operatorname{sign}(b_\ell^x[p]) \cdot W_{\ell-1}^x[p, q] x_{\ell-1}[q], \tag{14}$$

$$e(l-1, p, q) = |\operatorname{sign}(W_{\ell-1}^x[p, q] x_{\ell-1}[q])|. \tag{15}$$

We use the logistic function to attribute the bias so that the sum over all components recovers the original bias, and $T$ serves as a temperature parameter to control the sharpness of the attribution. With $T$ large, the bias is attributed in a more balanced manner, while with $T$ small, the bias is attributed mostly to a few neurons in layer $\ell - 1$. Also note that we only consider the non-zero components (indicator $\mathbb{1}_{e(l-1,p,q)=1}$ checks whether the component is zero), as the zero-valued components do not offer any contribution to the output value. For example, consider a convolutional layer, the corresponding matrix form is very sparse, and only the non-zero entries are involved in the convolution computation with a filter.

The second option adopts a different idea but with a similar philosophy of compensating the attribution of the gradient term. Again, the target is to achieve the value of $x_\ell[p]$, and we may assume that to achieve such a value, every component $W_{\ell-1}^x[p, r] x_{\ell-1}[r]$ should have an equal responsibility, which is the average target value, i.e., $x_\ell[p] / \sum_{r=1}^{d_{l-1}} \mathbb{1}_{e(l-1,p,q)=1}$ (again, we only need to consider

the contribution from non-zero components). The offsets of each component to the average target value can be treated as the causal factor of introducing the bias term, where components that are far from the average target get more compensation from the bias, and the components that are close to the target, since they already fulfill their responsibility, receive less compensation from the bias. This produces the following method to compute $s_\ell[p,q]$, i.e.,

$$s_\ell[p,q] = \left| \frac{x_\ell[p]}{\sum_{r=1}^{d_{l-1}} \mathbb{1}_{e(l-1,p,q)=1}} - W_{\ell-1}^x[p,q]x_{\ell-1}[q] \right|. \tag{16}$$

Note we use the same equations for $e(l-1,p,q)$ and $\alpha_l[p,q]$ as defined in Eq. (13)-(15)

The above two options are our designs for the $\alpha_\ell[p,q]$ function. The attribution function is valid as long as $\sum_{r=1}^{d_{l-1}} \alpha_\ell[p,r] = 1$. While we utilize the logistic function so that all attribution factors are positive, $\alpha_\ell[p,r]$ can be negative and still applicable to our BBp framework. We note that there is no single solution to get the optimal attribution function. The proposed options can be applied to any piecewise-linear deep neural networks, and for specific activation function, it is possible to design specialized attribution functions to get still better bias attribution.

## 5 EXPERIMENTS

### 5.1 IMPORTANCE OF BIAS IN DNNS

We first evaluate the importance of bias terms, or in other words, the amount of information encoded in the bias terms by comparing networks trained both with and without bias.

In Table 1, we compare results on the CIFAR-10 ,CIFAR-100 (Krizhevsky & Hinton, 2009) and Fashion MNIST (Xiao et al., 2017) datasets. We trained using the VGG-11 Network of (Simonyan & Zisserman, 2014), and we compare the results trained with bias, and without bias. Moreover, in the trained with bias case, we derive the linear model of every data point $g(x) = wx + b$, and compare the performance using only the resulting gradient term $wx$ and only the resulting bias term $b$. From the results shown in the table, we find that the bias term carries appreciable information and makes unignorable contributions to the correct prediction of the network.

Table 1: Compare the performance (in test accuracy %) of models with/without the bias terms. The "only $wx$" and "only $b$"columns use the same model as the "train with bias" column.

| Dataset | Train Without Bias | Train With Bias, Test All | Test Only $wx$ | Test Only $b$ |
|---|---|---|---|---|
| CIFAR10 | 87.0 | 90.9 | 71.5 | 62.2 |
| CIFAR100 | 62.8 | 66.8 | 40.3 | 36.5 |
| FMNIST | 94.1 | 94.7 | 76.1 | 24.6 |

### 5.2 BIAS ATTRIBUTION ANALYSIS VISUALIZATION

We present our bias attribution results, using the two options of attribution scores discussed in section 4.2, and compare to the attribution result based only on gradient information. We test BBp on STL-10 (Coates et al., 2011) and ImageNet(ILSVRC2012) (Russakovsky et al., 2015) and show the results in Fig. 2. For STL-10, we use a 10 layer convolutional network ((32,3,1),maxpool,(64,3,1),maxpool,(64,3,1),maxpool,(128,3,1),maxpool,(128,3,1),dense10, and (32,3,1) corresponds to a convolutional layer with 32 channels, kernel size 3 and padding 1), and for ImageNet we use the VGG-11 network of Simonyan & Zisserman (2014). For both gradient and bias attribution, we select the gradient/bias corresponding to the predicted class (i.e., one row for the final layer weight matrix and one scalar for the final layer bias vector). Note that BBp is a general framework and can work with other choices of gradients and biases for the last layer (e.g. the top predicted class minus the second predicted class).

From Fig. 2, we can observe that the bias attribution can highlight meaningful features in the input data, and in many cases, capture the information that is not covered by the gradient attribution. For example, for the "bird" image of STL-10, the bias attribution shows the major characteristics of a bird such as the body shape, the round head the yellow beak and the claws. Compared to the gradient attribution, the bias attribution seems to give cleaner explanations, and show much stronger focus on the head and beak. Overall, the bias attribution method tends to attribute in a more concentrated way, and often shows explanations about he DNNs complementary to the gradient information.

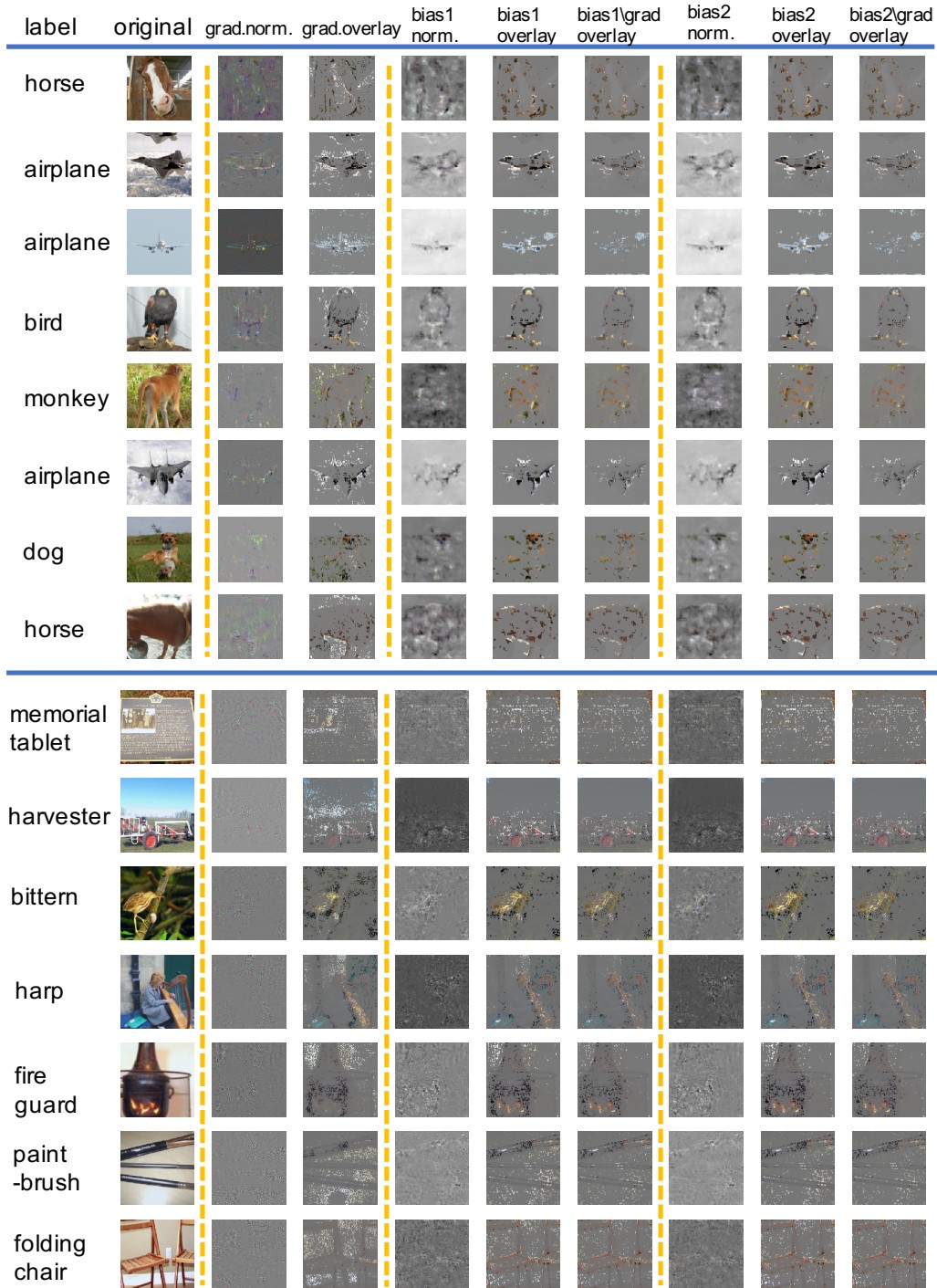

Figure 2: Bias attribution (BBp) on the STL-10 (top) and ImageNet (bottom) datasets compared to gradient-only attribution. The label of every image is shown in the leftmost column. The gradient attribution is the element-wise product between the linear model weight $w$ and data point $x$. The "grad.norm." and "bias norm." columns show the attribution of gradient and bias normalized to the color range (0-255). The "grad.overlay" and "bias.overlay" show the 10% data features with the highest attribution magnitude of the original image. "bias\grad overlay" columns show the data features selected by the bias attribution and not selected by the gradient attribution. Bias1 correspond to the first proposed option of calculating the bias attribution score (Eq. (13)-(15)), and bias2 (Eq. (16)) correspond to the second proposed option. For both options of calculating the bias attribution score, the temperature parameter $T$ is set to 1. For many cases, the bias attributions offer explanations complementary to the information provided by the gradient. For example, for the "bittern" image from ImageNet, BBp shows stronger attribution on the bird's head and body compared to the gradient method. For the "fire guard" image of ImageNet, BBp has clear attribution to the fire, in addition to the shape of the guard, while the gradient method only shows the shape of the guard. Similarly, for the "folding chair" of ImageNet, BBp shows clearer parts of the chair, while the gradient attribution shows less relevant features such as the background wall.

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
