# OpenReview forum: "Bias Also Matters: Bias Attribution for Deep Neural Network Explanation"
_ICLR.cc/2019/Conference_

### Official Review · AnonReviewer1 · 2018-11-01
**interesting direction; preliminary result**

**Rating:** 5
**Confidence:** 4

**Review:**

The paper starts by establishing that biases play an important, negligible role in existing DNNs.
Specifically, they help improve classification performance, and networks trained with biases do make use of biases.

Then, the authors recognize that the state of the art DNNs use ReLU and variants, which are a piece-wise linear function.
Over the linear regions, the entire DNN can be collapsed into a single linear model f(x) = Wx + b.

Then the authors argue that the existing gradient-based attribution methods (for interpreting DNNs) often ignore the attribution of the `b` terms in the heatmap.
That is, when backpropagating the DNN outputs back to the input, the gradient of (Wx + b) wrt x is exactly W only (ignoring the contribution of b).

The paper then proposes a method for backpropagating biases.
From the presented results, I only can conclude that bias backpropagation does show a different heatmap compared to regular gradient-based methods.
However, it is unclear how much this BBp result is advancing our understanding of DNNs.
The result for this is still preliminary.

- Clarity
Research is well motivated, and paper presentation shows a nice, coherent story.

- Originality
AFAIK, the direction of looking at bias attribution is novel.

- Significance
The significance of the paper is limited because (1) the paper only considers the positive region of ReLUs; (2) the empirical results are preliminary and do not show a convincing usefulness of BBp.
Suggestions: authors may design a toy dataset or find a dataset that has some inherent biases (e.g. data imbalance) to show that DNNs do capture interesting information in the biases. From there, hopefully the impact of BBp can be clearer.

At the moment, the paper appears not ready for publication.

---

### Official Review · AnonReviewer3 · 2018-11-02
**Interesting work, but needs refinement.**

**Rating:** 5
**Confidence:** 5

**Review:**

Summary of the paper
This paper proposes a method for attributing the output of a neural network to bias terms. The method is restricted to networks that have piecewise-linear activation functions. Computation is recursive, starting from bias attribution to the activation of the penultimate layer, such that the final attribution is of the same size as the input data point and sums up to the bias term when the network is written as a linear function (for that input).

Strengths
- The idea of mapping the bias term back to the input is interesting as it shows a common behaviour of the network on inputs that choose the same pieces of the piecewise linear functions.
- Other gradient-based methods overlook the bias term when piecewise-linear activations are involved, so this method closes that gap.

Questions for authors
- The separability of the bias and gradient terms is possible only for piecewise linear activation functions, and would not generalize to other activations (e.g., LSTMs in NLP).
- Except for the 1-2 examples pointed out by the authors, it is not clear from the visualizations that bias attribution shows something qualitatively different. For instance, in “airplane”, “horse” and “fireguard”, gradient also highlights a region similar to bias attributions (although, technically they are complementary).
- While the authors qualitatively compare with other attribution methods, they only experimentally compare with gradient. It would be instructive to compare with more refined gradient-based attribution methods such as Integrated Gradients or DeepLift and show empirically that looking at bias attribution is better over simply looking at other attribution-based methods. Integrated Gradients specifically argues that it removes extraneous attribution to background (http://www.unofficialgoogledatascience.com/2017/03/attributing-deep-networks-prediction-to.html).
- To have substantial content for a full publication, it may be good to address what insights one can derive from bias attributions. For instance, cluster inputs based on bias attributions, and show how different sets of inputs may be affected by different kinds of biases. For e.g., do all images of birds show similar bias attributions? Or, for instance, does a picture of a house (which may have the same kind of edges as a chair) have the same bias attribution as that of a chair?

Conclusion
I think the paper is interesting, but for a full publication, a thorough comparison with other methods is required, as well as showing more insights as to how bias attribution is useful (another e.g., can it be combined with gradient-attribution for a “richer” visualization?)

---

### Official Review · AnonReviewer2 · 2018-11-15
**Interesting; Lack of comparisons to some existing methods.**

**Rating:** 5
**Confidence:** 5

**Review:**

Summary:
The paper's main contribution is to attribute the bias term seen at the output to each of the input dimension appropriately.  The other claim is that together with gradient information , this could enhance existing explainability methods.

The paper considers DNNs with piecewise linear activation functions. Then the final DNN output is a piece linear function. So for any given point x, the point lies in one of the linear pieces and there fore, can be written as a linear model. The gradient term can be computed using back propagation methods (although back propagating keeping the weights fixed and keeping the input as the variable). However there are no know existing works that attribute the bias of the linear piece at the final layer to the input dimensions. This paper provides a method to do it such that when you add the vector contribution of all the dimension in the input - it results in the bias vector at the output layer.

The basic idea is to distribute dimension wise bias attribution  at layer \ell to layer \ell-1 by using N_{\ell} x N_{\ell-1} attribution matrix where N_{\ell} is the dimension of layer \ell. This is done using two methods - one using exponential weights and the other using some sort of variance measure from a fixed average bias.

The authors then show using examples from STL-10 and Imagenet datasets, how the gradients and biases attributed to the inputs compare for explanation purposes.

Strengths:

The notion of attributing final layer biases to input layer is novel. Its important given that it carries important information regarding the final classification output.

Weaknesses:

a) This paper lacks quite a bit on comparison with existing work. For example, LRP (layer wise relevance propagation) has been referred to by the authors. However, there is no comparison with LRP heat maps. This website - http://www.explain-ai.org/ - documents state of the LRP methods with code, videos, papers (some of which have been cited by the authors). I think the authors could produce heat map produced by LRP for all these different examples in page 8. For instance, pls look at Image A in Fig .2 in https://arxiv.org/pdf/1708.08296.pdf. There is 'cup' and a 'volcano' in the same picture. The paper compares gradient based heatmaps and LRP based ones. LRP based ones are very crisp (for this image of course). LRPcode is readily available from a well maintained project page - http://www.explain-ai.org/. Will the heat maps of grad/bias look crisper than LRP ? Other papers and more examples of LRP can be found at http://www.explain-ai.org/

LRP takes the final probability weight at the output layer and assigns recursively to other neurons in the penultimate very similar to what the current paper does for bias. However, the attribution mechanism (the weights) are different. A couple of variations are explored in this survey (https://arxiv.org/pdf/1708.08296.pdf).

b)  This point is related to the first- The authors say "Therefore, the interpretation of the DNN’s behavior on the input data should be exclusive to the information embedded in the linear model." - I disagree a little bit here. It is true that behavior of the DNN on the input is exclusive to the linear piece. However, interpretation of the behavior/ explanation of it is another matter. For example, I quote two methods in the literature that have been used for explanation of an input sample - but does not use the linear piece or the gradient information.
     1)  Pls look at - https://arxiv.org/pdf/1703.02647.pdf - (Figure 3 + Section A.8 in the appendix). The paper used streaming submodularity algorithms to actually assemble a part of the image with everything else "blacked out" to determine which sparse parts of the image are responsible for the final output. They have exhaustive comparisons with LIME too. These methods actually rely on behavior of DNN far away from the actual x to explain the behavior at x. "Zeroing out irrelevant" parts is one of the ways of explaining adopted by LIME and these approaches. Ideally the authors should compare with these too.

 2) In this paper - https://arxiv.org/abs/1802.07623 - authors provide pertinent negatives - what should "not be" there in the image so that the label does not change - In fact for MNIST data, this produces very interesting additional explanations that is not produced by methods (including LRP and LIME) that rely on things in the image. Again the explanation is not at all related to the linear piece.

In general, linear pieces are so close by , the nearby movement and possibly change of gradients can also provide useful explanation of the behavior.


Overall - at least the authors must discuss the above references + also survey works that highlights "relevant parts of the image" like LIME and streaming sub modularity etc. Further an actual comparison to LRP (code is easily available) is crucial to evaluate the efficacy of the proposed methods given that the authors of LRP have compared with gradient based methods. Comparison with LIME would also be interesting and desirable.

---

### Meta-Review · Area_Chair1 · 2018-12-12
**Method to visualize spatial bias distribution in network. Reviewers unanimous reject, no rebuttal from authors.**

**Confidence:** 4
**Recommendation:** Reject

**Metareview:**

The work presents a method to back propagate and visualize bias distribution in network as a form of explainability of network decisions. Reviewers unanimous reject, no rebuttal from authors.